# Effects of Overload on Thermal Decomposition Kinetics of Cross-Linked Polyethylene Copper Wires

**DOI:** 10.3390/polym15193954

**Published:** 2023-09-30

**Authors:** Yizhuo Jia, Pengrui Man, Xinyao Guo, Liang Deng, Yang Li

**Affiliations:** Forensic Science Institute, China People’s Police University, Langfang 065000, China; jiayizhuo@126.com (Y.J.); mmm2311592503@163.com (P.M.); 15800983723@163.com (X.G.)

**Keywords:** overload fault, cross-linked polyethylene copper wire, thermal decomposition, reaction mechanism, functional group

## Abstract

During an overload fault in an energized wire, the hot metal core modifies the structure of the insulation material. Therefore, understanding the thermal decomposition kinetics of the insulation materials of the overloaded wire is essential for fire prevention and control. This study investigates the thermal decomposition process of new and overloaded cross-linked polyethylene (XLPE) copper wires using thermogravimetry–Fourier-transform infrared spectroscopy and cone calorimetry. The thermal decomposition onset temperature and activation energy of the overloaded XLPE insulation materials were reduced by approximately 15 K and 20 kJ mol^−1^, respectively, and its reaction mechanism function changed from D-ZLT_3_ to A2 (0 < α < 0.5). The FTIR shows that the major spectral components produced during the pyrolysis of the XLPE insulation material are C-H stretching, H_2_O, CO_2_, C-H scissor vibrations, and C=O and C=C stretching. Additionally, the four functional groups in the PE chains produced the spectral components in the following decreasing order of wavenumber: C–H stretching > CO_2_ > C–H scissor vibration > C=O and C=C stretching.

## 1. Introduction

In China, electrical fault fires account for approximately 30% of all fires, of which 50% are caused by burning wires and cables [1]. With the increasing electrification of society, an increasing number of electrical appliances and equipment are connected to electrical circuits. When the wire diameter in an electrical circuit is insufficient or the overcurrent protection device fails, the current passing through the wire surpasses its safe limit, which is referred to as an overload fault in the Guide for Fire and Explosion Investigations (NFPA 921-2021) [2]. Exceeding the safe current limit for a prolonged period of time causes the Joule heat generated by the metal core of the wire to damage the internal structure of the polymer insulation via thermal conduction, thereby altering fire hazard [3,4]. Cross-linked polyethylene (XLPE)-insulated copper wires with good electrical and mechanical properties are widely used in nuclear power plants, hospitals, schools, and residential buildings. XLPE is flammable and produces numerous hydrocarbons at high temperatures after combustion. The initial phase of polymer combustion is thermal decomposition and is important in assessing the fire hazard of a material [5,6,7,8,9,10].

Previous studies have investigated the thermal mass loss process, reaction mechanism, and gas evolution products of various PE insulation materials using thermogravimetric analysis (TGA) and Fourier-transform infrared spectroscopy (FTIR). Cho et al. [11] investigated the thermal decomposition kinetics of low-density PE (LDPE) and XLPE using TGA and the Kissinger–Akahira–Sunose (KAS) equation. They found that the activation energies of LDPE and XLPE were 146.8 and 170.4 kJ mol^−1^, respectively. Park et al. [12] studied the thermal decomposition of high-density PE (HDPE), LDPE, and linear LDPE at a temperature increase rate of 10–50 K/min and found their activation energies to be 333.2–343.2, 187.5–199.1, and 219.2–230.1 kJ mol^−1^, respectively. Aboulkas et al. [13] performed TGA of HDPE and LDPE at different heating rates (2, 10, 20, and 50 K/min) in the temperature range of 300–900 K under a nitrogen atmosphere. They found that the pyrolysis reaction models of HDPE and LDPE can be described using the R2 model, with activation energies of 238–247 and 215–221 kJ mol^−1^, respectively. Mo et al. [14] studied the pyrolysis of XLPE insulation materials inside a cable using TGA–DSC and showed that the materials mainly underwent a significant loss of mass. Wang et al. [15] investigated the thermal decomposition behavior of XLPE sheaths in flame-retardant high-voltage cables using thermogravimetry–Fourier-transform infrared spectroscopy (TG–FTIR) and found that the first-order model (F1) played an important role in the pyrolysis process. The main peaks of the FTIR spectrum of the pyrolyzed XLPE corresponded to C–H stretching, C–H scissor vibration, CO_2_, and C=O and C=C stretching. Sugimoto et al. [16] analyzed the oxidation products of XLPE during thermal radiation aging using FTIR and found that carboxylic acids, carboxylic acid esters, and carboxylic acid anhydrides were the major products. Wang et al. [17] studied the thermal decomposition process of new and aged LDPE insulation materials using TGA and analyzed the pyrolysis reaction model using generalized master diagram method. They found that the “contracting area” (R2) model is more suitable for the thermal decomposition process of new and aged LDPE insulation materials.

Previous studies have mainly focused on the effects of different insulation types and external environmental conditions on the thermal decomposition of their insulation materials. However, limited research has been conducted on variations in the thermal decomposition kinetics of their XLPE insulation materials is scarce. Therefore, in this study, the thermal decomposition kinetics of new and overloaded XLPE insulation materials of the wires were investigated in detail using TG-FTIR. Model-free and model-fitting methods were used to evaluate apparent activation energies and reaction mechanisms. The FTIR studies provided information on absorbance changes as a function of wavenumber and temperature, which allowed the identification of gases released during the combustion of new and overloaded XLPE insulation materials. The results of this study have important implications for wire insulation improvement, fire risk assessment, pyrolysis modeling, and pyrolysis reuse.

## 2. Materials and Methods

### 2.1. Materials

The outer diameter, insulation thickness, and safe current value of the XLPE-insulated copper wire used in this study were 0.90 mm, 0.80 mm, and 25 A, respectively. The wire was manufactured by China Far Eastern Cable Group Co. (Yixing, China). The main components of the polymeric insulation layer were XLPE and additives, and no flame retardant was added to the insulation layer. The samples conformed to Chinese standard GB/T 19666-2019 [18] and IEC 60331-2 [19]. The specific specification parameters of the wires are shown in Table 1. Prior to testing, all samples were dried at 60 °C for 24 h to remove any moisture.

### 2.2. Experimental Method

An XLPE copper wire was prepared after an overload failure, and then a new XLPE copper wire was selected as a control sample. TG-FTIR was used to investigate the thermal decomposition processes of the new and overloaded XLPE insulation materials. Three parallel experiments were conducted for each group to ensure the accuracy of the experimental results.

#### 2.2.1. Preparation of Overloaded XLPE Copper Wires

According to the U.S. Handbook of Fire Protection Engineering [20], when the current flowing through a wire exceeds its safe current value (I_e_) by three times, the Joule heat generated by the internal metal core destroys the physical and chemical structure of the insulation. Furthermore, oxidation discoloration occurs on the outer surface of the insulation material. Such overload faults do not affect the normal operation of electrical wires in the short term and are therefore difficult to detect; however, they pose a substantial risk of fire. Therefore, in this study, a constant DC current of 80 A (3.2I_e_) was passed through the XLPE copper wire for 1 h. This time was sufficient to significantly damage the insulation material due to the Joule heat generated by the metal core.

#### 2.2.2. TG–FTIR Apparatus

Thermal analysis of the new and overloaded XLPE insulation materials was performed using a STA449 F5 thermogravimeter (Netzsch, Selb, Germany) coupled with a Bruker Invenio S spectrometer (Bruker, Selb, Bavaria, Germany). For each experiment, the XLPE insulation material was stripped from the wire using wire strippers to obtain test samples weighing approximately 5 mg in an alumina crucible. The experiments were performed under the nitrogen atmosphere at the heating rates of 5, 10, 20, 30, and 40 K min^−1^ and the experimental temperatures of 308–1273 K. To ensure experimental accuracy and prevent condensation of volatile pyrolysis gas released from the XLPE insulation materials during heating, the temperature of the transmission line between the TG and the FTIR spectrometer was set to 473 K [21]. The FTIR spectra were recorded with a scanning frequency of 32 min^−1^ in the spectral range of 4000 to 650 cm^−1^ and a resolution of 4 cm^−1^. All interferograms were acquired during pyrolysis to obtain the absorption spectra and corresponding evolution gas spectra. The purge flow rate of the spectrometer was 50 mL min^−1^.

## 3. Results and Discussion

### 3.1. Thermogravimetric Loss

Figure 1 and Figure 2 show the TG and DTG profiles, respectively, of new and overloaded XLPE insulation materials under a nitrogen atmosphere. The TG and DTG profiles show the total mass loss and mass loss rate, respectively, as a function of temperature at different heating rates of 5, 10, 20, 30, and 40 K min^−1^. All the TG and DTG curves exhibited a similar trend, indicating that overload faults do not significantly alter the basic reaction mechanism of XLPE insulation materials during thermal decomposition. The thermal decomposition kinetics of XLPE insulation materials is generally considered to be a one-step process [15]. This is confirmed by the presence of only one mass loss phase and one peak in the TG and DTG curves, respectively. During thermal decomposition, the chemical structure of the insulation layer undergoes significant changes, including the cleavage and generation of free radicals. As seen in Figure 2a, the XLPE insulation materials tend to slightly lose mass at approximately 400 K, probably because of the removal of brittle additives, such as CaCO_3_. In the temperature range of 500–700 K, the molecular chains tend to break, releasing alkyl radicals (–CH_3_). When the temperature is increased to 700–800 K, the molecules further cleave and form abundant hydrocarbon gases [22], such as methane, ethylene, and other flammable gases, which is the main process of the thermal decomposition of XLPE insulation materials. With the further increase in the temperature, the XLPE insulation materials continue to decompose and carbonize. Notably, the TG and DTG curves shifted to higher temperatures as the heating rate increased. This phenomenon has been described and explained by various researchers [12,23].

Table 2 lists the thermal decomposition parameters of the new and overloaded XLPE insulation materials. The thermal decomposition onset temperature (T_onset_), peak DTG curve (DTG_peak_), and maximum temperature required to reach the DTG_peak_ (T_peak_) of the overloaded XLPE materials were lower than those of the new XLPE materials, regardless of the heating rate. These results suggest that owing to the overload fault in the XLPE copper wires, the insulation becomes more susceptible to thermal decomposition, and the reaction rate during thermal decomposition is decreased. When an overload fault occurs, the heat generated by the metal core alters the molecular structure of the XLPE insulation by breaking and rearranging its molecular chains. Due to the occurrence of overload faults, carbonization of the outer surface of the insulation occurs and therefore T_onset_ rises. This could be the reason for the difference in thermal decomposition behavior of the new and overloaded XLPE-insulated copper wires.

### 3.2. Activation Energy

The activation energy calculated in this study is the energy required to break the internal molecular bonds of a material during thermal decomposition. It is an important parameter for assessing the thermal stability of a material. Model-free methods have been widely and commonly used for estimating the activation energy of composites, without requiring any prior knowledge or assumptions about a reaction model. Two of the most commonly used model-free methods are KAS and Flynn–Wall–Ozawa (FWO) methods [24,25]:(1)$$\ln\left(\frac{\beta }{T_{\alpha }^{2}}\right)=\ln\frac{A_{\alpha }R}{E_{\alpha }G\left(\alpha \right)}-\frac{E_{\alpha }}{RT_{\alpha }}$$
(2)$$ln\beta =ln\frac{A_{\alpha }E_{\alpha }}{RG\left(\alpha \right)}-5.331-1.052\frac{E_{\alpha }}{RT_{\alpha }}$$
where β is the heating rate (K s^−1^); α is the degree of conversion; $A_{\alpha }$, $E_{\alpha },$ and $T_{\alpha }$ represent the distributed the pre-exponential factor (s^−1^), activation energy (kJ mol^−1^), and temperature (K) with the degree of conversion α, respectively; R is the universal gas constant (8.314 kJ^−1^ mol^−1^); and G(α) is the integral form of the reaction model.

Table 3 and Table 4 list the specific values calculated using the KAS and FWO methods, respectively. The activation energies of the new and overloaded XLPE insulation materials increased and then decreased with the increasing conversion rate. The average values of activation energies of the new XLPE insulation materials computed using the KAS and FWO methods were 293.06 and 290.65 kJ mol^−1^, while those of the overloaded XLPE insulation materials were 272.29 and 270.75 kJ mol^−1^, respectively. The slight fluctuations in the activation energies estimated by the KAS and FWO methods were attributed to differences in the approximations of the two model-free methods. 

Figure 3 shows the change in activation energy for new, overloaded XLPE insulation at different conversion rates. The activation energy value of the overloaded XLPE insulation materials is lower than that of the new XLPE insulation materials, implying that the thermal stability of the XLPE insulation materials decreased due to the overload fault. This observation is consistent with the T_onset_ results of the new and overloaded XLPE insulation materials discussed in Section 3.1. Overloading of XLPE insulation materials leads to a reduction in their activation energy. This is because, during an overload fault, the elevated temperature of the metal core weakens the intermolecular interaction force within the insulation layer. Furthermore, the high temperature of the metallic copper core results in some of the copper metal penetrating into the XLPE insulation materials. This infiltration can catalyze or activate the thermal decomposition reaction of XLPE, facilitating its decomposition and weight loss. The generation of macromolecular radicals, perhydrates, and heat-stable products (ketones, alcohols, and acids) occurs.

### 3.3. Reaction Mechanism Function

The reaction mechanism function is a model to calculate the chemical reaction rate and predict product generation of materials. Furthermore, this model can predict the pyrolysis behavior of materials and, to some extent, can reflect their combustion process. The Málek method is a better approach for determining f(α) or $G\left(\alpha \right)$ by defining the function y(α), which can be expressed as [26,27]
(3)$$y(\alpha )={\left(\frac{T}{T_{0.5}}\right)}^{2}\frac{\left(\frac{d\alpha }{dt}\right)}{{\left(\frac{d\alpha }{dt}\right)}_{0.5}}=\frac{f(\alpha )\cdot G(\alpha )}{f(0.5)\cdot G(0.5)}$$
where the corresponding temperature *T* can be substituted at the selected α under different β, $y(\alpha )={\left(T/T_{0.5}\right)}^{2}\left(d\alpha /dt\right)/{\left(d\alpha /dt\right)}_{0.5}$ can be obtained from the experimental data, and the y(α)=f(α)⋅G(α)/f(0.5)⋅G(0.5) can be obtained as a standard curve. Thus, comparison between the experimental and the standard curves is similar to the corresponding mechanism function. The common reaction mechanisms function of solid-state reactions are provided in Table 5 [28].

Figure 4 shows the experimental (scatter) and theoretical (line) plots of y(α) at different heating rates of the new and overloaded XLPE insulation materials. Different color realizations represent different reaction mechanisms. The overall reaction mechanisms of the new XLPE insulation materials were the closest to those of D-ZLT_3_. However, the overall reaction mechanisms tended toward A2 (0 < α < 0.5) and D-ZLT_3_ (0.5 < α < 1) for the overloaded XLPE insulation materials. From Section 3.1, it is clear that the Joule heat generated in the metallic copper core during an overload fault can damage the internal structure and chemical composition of the XLPE insulation materials. These changes will change the course of thermal decomposition; therefore, the reaction mechanism functions of the overloaded and new XLPE insulation materials will differ slightly.

### 3.4. FTIR Analysis

The products evolved during the thermal decomposition of materials were analyzed through FTIR spectroscopy. The three-dimensional spectra of the new and overloaded XLPE insulation materials can provide important information, such as the type of the gas and its concentration as functions of temperature and wavenumber. Typical three-dimensional spectrograms of the new and overloaded XLPE insulation materials are shown in Figure 5. The gas temperatures generated by the new and overloaded XLPE insulation materials ranged between 750 K and 850 K. These values are in good agreement with the evolution of the DTG curves. It is noteworthy that the temperature corresponding to the spectral intensity peak is determined by TG, as there is a delay of several seconds between TG and FTIR [29].

Figure 6 shows the FTIR spectra of the new and overloaded XLPE insulation materials at a heating rate of 40 K min^−1^. For the XLPE insulation materials, the absorption bands at 4000–3500 cm^−1^ correspond to the presence of H_2_O produced due to the decomposition of oxygen-containing groups [30]. The signals at around 2928 cm^−1^ and 2860 cm^−1^ are associated with the symmetric and asymmetric stretching vibrations of C–H bonds, respectively, in –CH_2_ and –CH_3_ end groups present in PE chains. [31] Moreover, a prominent spectral band at 2400–2260 cm^−1^ corresponded to the presence of CO_2_ [32]. The spectral bands in the range of 1750–1652 cm^−1^ were attributed to the stretching vibrations of unconjugated C=O and C=C bonds [33]. The bands around 1460 cm^−1^ and 732 cm^−1^ corresponded to the scissor vibrations of the C–H bonds in the CH_2_ groups in the aliphatic chains [34,35,36,37,38]. Thus, the main spectral components produced during the pyrolysis of the XLPE insulation material are C–H stretching, H_2_O, CO_2_, C–H scissor vibrations, and C=O and C=C stretching.

Figure 7 shows the main bond change curves during pyrolysis of new and overloaded XLPE insulation materials at a heating rate of 40 K min^−1^. Notably, the concentration of gases released during the thermal decomposition of the overloaded XLPE insulation materials was higher than that of the new XLPE insulation materials. Additionally, the four functional groups in the PE chains produced the spectral components in the following decreasing order of wavenumber: C–H stretching > CO_2_ > C–H scissor vibration > C=O and C=C stretching.

## 4. Conclusions

In this study, the insulation materials of new and overloaded XLPE copper wires were analyzed using TG–FTIR experiments. The activation energies at different conversion rates were estimated using a model-free method, and the reaction mechanism was predicted using the Málek model. The main components of the gases evolved during thermal decomposition were investigated using FTIR spectroscopy. The key results can be summarized as follows:Both the new and overloaded XLPE insulation materials underwent thermal decomposition in one step, but exhibited different thermal decomposition behaviors. The T_onset_, DTG_peak_, T_peak_, and residual mass of the overloaded XLPE insulation materials were lower than those of the new XLPE insulation materials, regardless of their heating rates.The activation energies of the overloaded XLPE insulation materials, calculated using the KAS and FWO model-free methods, were lower than those of the new XLPE materials. For the new XLPE insulation materials, the overall reaction mechanism was more consistent with that of the D-ZLT_3_ model. However, for the overloaded XLPE insulation materials, the reaction mechanism corresponds to A2 (0 < α < 0.5) and D-ZLT_3_ (0.5 < α < 1).The results of FTIR analysis showed that the main spectral components of the pyrolysis gases of the new and overloaded XLPE insulation materials were C–H stretching, H_2_O, CO_2_, C–H scissor vibrations, and C=O and C=C stretching. The concentrations of gases released from the overloaded XLPE insulation materials were lower than those released from the new XLPE insulation materials. The amount of the four spectral components produced during thermal decomposition was in the following order: C–H stretching > CO_2_ > C-H scissor vibration > C=O and C=C stretching.

## Figures and Tables

**Figure 1 polymers-15-03954-f001:**
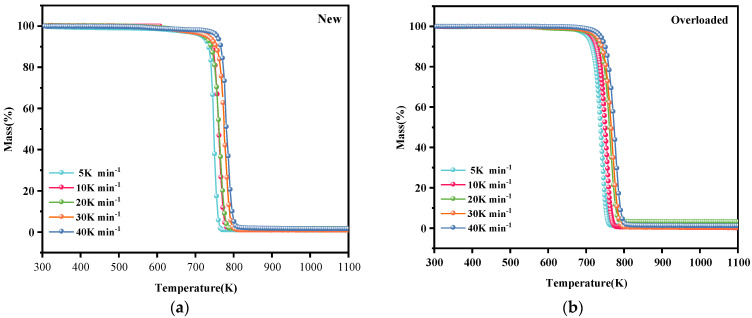
Total mass loss (TG) profiles of (**a**) new and (**b**) overloaded XLPE insulation materials at different heating rates under a nitrogen atmosphere.

**Figure 2 polymers-15-03954-f002:**
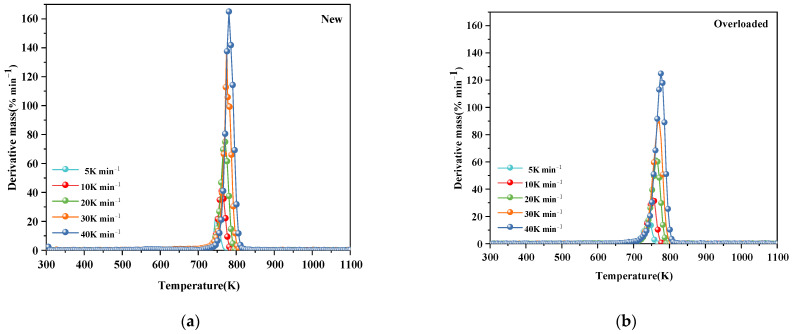
Mass-loss rate (DTG) profiles of (**a**) new and (**b**) overloaded XLPLE insulation materials at different heating rates under a nitrogen atmosphere.

**Figure 3 polymers-15-03954-f003:**
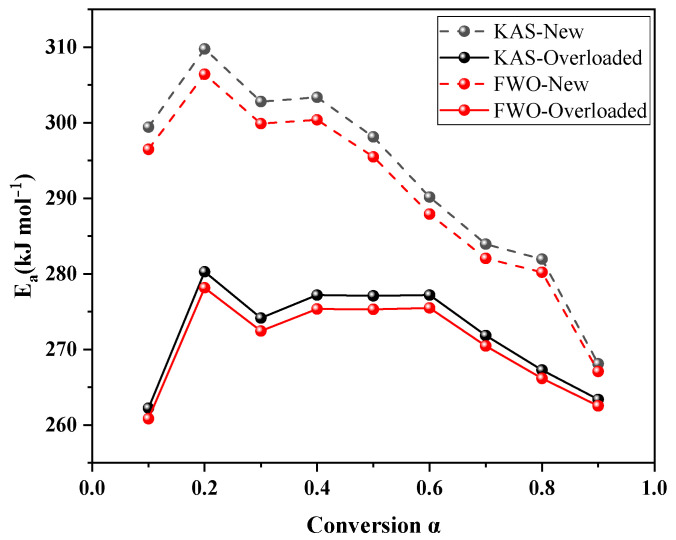
Activation energy at different conversion α.

**Figure 4 polymers-15-03954-f004:**
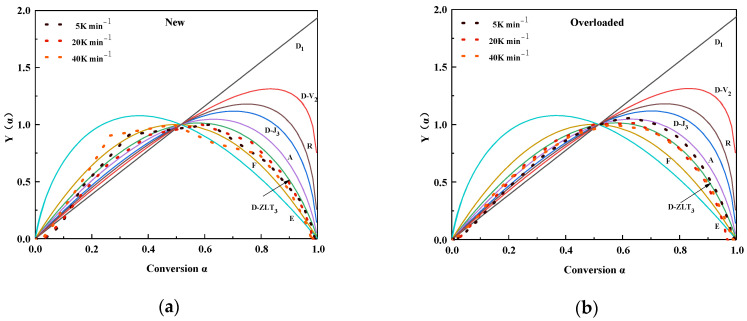
Experimental and standard *y*(α) plots of (**a**) new and (**b**) overloaded XLPE insulation materials calculated at different heating rates.

**Figure 5 polymers-15-03954-f005:**
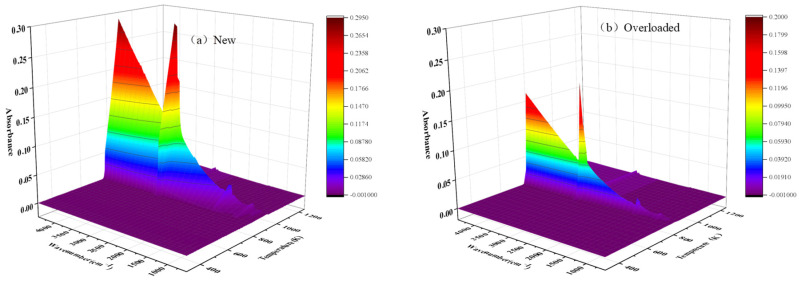
Three-dimensional TG–FTIR spectrograms of the (**a**) new and (**b**) overloaded XLPE-insulated copper wires that were heated at a heating rate of 40 K min^−1^.

**Figure 6 polymers-15-03954-f006:**
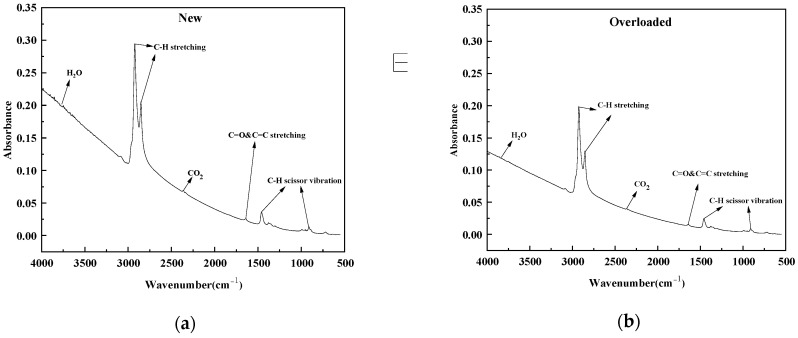
Absorption spectra of (**a**) new and (**b**) overloaded XLPE insulation at the heating rate materials of 40 K min^−1^.

**Figure 7 polymers-15-03954-f007:**
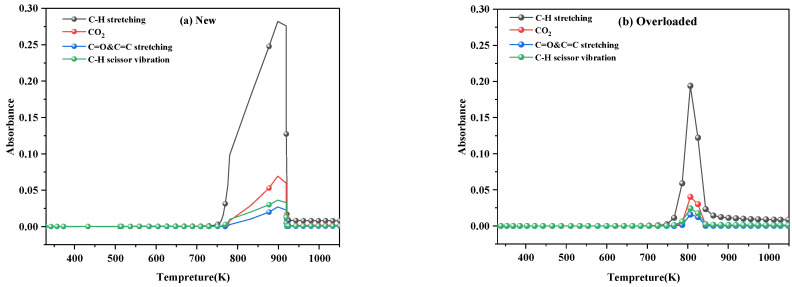
Release profiles of the main bonds of the (**a**) new and (**b**) overloaded XLPE insulation materials at a heating rate of 40 K min^−1^.

**Table 1 polymers-15-03954-t001:** Physical properties of the XLPE-insulated copper wires.

Characteristic	Cu (99.99%)	Insulation (XLPE)
Diameter (mm)	0.16 ± 0.01	0.8 ± 0.02
Density (g cm^−3^)	8.9 ± 0.01	0.93 ± 0.01
Heat conductivity (W m^−1^ K^−1^)	400 ± 5.00	0.42 ± 0.02
Resistivity (Ω m)	1.75 × 10^−8^	/
Specific heat (J g^−1^ K^−1^)	0.39 ± 0.01	1.9 ± 0.01
Melting point (K)	1357.77 ± 20.0	383–393

**Table 2 polymers-15-03954-t002:** Thermal decomposition parameters of new and overloaded XLPE insulation materials.

Sample	Heating Rate/min^−1^	T_onset_/K	DTG_peak_/(% min^−1^)	T_peak_/K
New	5	739.9	23.0	745.1
	10	749.6	41.2	761.7
	20	754.8	74.9	770.7
	30	771.5	119.8	776.3
	40	776.9	165.0	780.7
Overloaded	5	723.1	17.0	741.9
	10	735.1	33.7	754.3
	20	744.8	61.1	761.6
	30	748.4	89.9	771.4
	40	755.8	124.9	775.8

**Table 3 polymers-15-03954-t003:** Activation energies obtained using the KAS method.

XLPE Insulation Materials	Conversion Rate	$E_{\alpha }$ (kJ mol^−1^)	R^2^
New	0.1	299.4	0.86
	0.2	309.7	0.95
	0.3	302.7	0.98
	0.4	303.3	0.98
	0.5	298.1	0.99
	0.6	290.1	0.99
	0.7	283.9	0.99
	0.8	281.9	0.99
	0.9	268.1	0.99
Overloaded	0.1	262.2	0.98
	0.2	280.2	0.98
	0.3	274.1	0.98
	0.4	277.1	0.99
	0.5	277.0	0.99
	0.6	277.1	0.99
	0.7	271.8	0.98
	0.8	267.2	0.98
	0.9	263.3	0.99

**Table 4 polymers-15-03954-t004:** Activation energies obtained using the FWO method.

XLPE Insulation Materials	Conversion Rate	$E_{\alpha }$ (kJ mol^−1^)	R^2^
New	0.1	296.4	0.87
	0.2	306.4	0.95
	0.3	299.8	0.98
	0.4	300.3	0.98
	0.5	295.4	0.99
	0.6	287.9	0.99
	0.7	282.0	0.99
	0.8	280.2	0.99
	0.9	267.0	0.99
Overloaded	0.1	260.8	0.98
	0.2	278.1	0.98
	0.3	272.4	0.98
	0.4	275.3	0.99
	0.5	275.3	0.99
	0.6	275.4	0.99
	0.7	270.4	0.98
	0.8	266.1	0.98
	0.9	262.5	0.99

**Table 5 polymers-15-03954-t005:** Common reaction mechanisms of solid-state reactions.

Number	Model	Differential Form f(α)	Integral Form G(α)
Diffusion Model			
1	1D diffusion D_1_	$-1/2\alpha^{-1}$	$\alpha^{2}$
2	2D diffusion-Valensi D-V2	${\left[-\ln\left(1-\alpha \right)\right]}^{-1}$	$\alpha +\left(1-\alpha \right)\ln\left(1-\alpha \right)$
3	3D diffusion-Jander D-J3	$6{\left(1-\alpha \right)}^{2/3}{\left[1-{\left(1-\alpha \right)}^{1/3}\right]}^{1/2}$	${\left[1-{\left(1-\alpha \right)}^{1/3}\right]}^{1/2}$
4	3D Zhuravlev-Leskin-Tempelman D-ZLT_3_	$3/2{\left(1-\alpha \right)}^{4/3}{\left[{\left(1-\alpha \right)}^{-1/3}-1\right]}^{-1}$	${\left[{\left(1-\alpha \right)}^{-1/3}-1\right]}^{2}$
Sigmoidal rate equations			
5	Avarami–Erofeev A2	$1/2\left(1-\alpha \right){\left[-\ln\left(1-\alpha \right)\right]}^{-1}$	${\left[-\ln\left(1-\alpha \right)\right]}^{2}$
6	Avarami–Erofeev A3	$1/3\left(1-\alpha \right){\left[-\ln\left(1-\alpha \right)\right]}^{-2}$	${\left[-\ln\left(1-\alpha \right)\right]}^{3}$
7	Avarami–Erofeev A4	$1/4\left(1-\alpha \right){\left[-\ln\left(1-\alpha \right)\right]}^{-3}$	${\left[-\ln\left(1-\alpha \right)\right]}^{4}$
Reaction order models			
8	Second-order chemical reaction F2	${\left(1-\alpha \right)}^{2}$	${\left(1-\alpha \right)}^{-1}-1$
9	Third-order chemical reaction F3	${\left(1-\alpha \right)}^{3}$	$-1/2\left(1-{\left(1-\alpha \right)}^{-2}\right)$
Exponent power models			
10	First-order E1	α	lnα
11	Second-order E2	1/2α	$\ln\alpha^{2}$
Geometrical contraction models			
12	Contracting area R2	$2{\left(1-\alpha \right)}^{1/2}$	$1-{\left(1-\alpha \right)}^{1/2}$
13	3D contracting volume R3	${\left(1-\alpha \right)}^{2/3}$	$3\left[1-{\left(1-\alpha \right)}^{1/3}\right]$

## Data Availability

The data presented in this study are available on request from the corresponding author.

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
