# Peer review of "Effects of Overload on Thermal Decomposition Kinetics of Cross-Linked Polyethylene Copper Wires"

_polymers, 2023, doi:10.3390/polym15193954_

Round 1

Reviewer 1 Report

This paper is well written. I did not find any remarks that should be addressed. In my opinion this paper is suitable for publication in this form. 

Author Response

Dear Reviewer,

I would like to express my sincere gratitude for your valuable feedback and guidance during the review process of my article. Your professional insights and meticulous review work have been immensely helpful to me, providing invaluable direction and the opportunity to enhance my research.

In accordance with your suggestions, I have made revisions and improvements to the article. Here are the key modifications I have implemented:

Comments

Reviewer :

Responses to reviewer’s comments

This paper is well written. I did not find any remarks that should be addressed. In my opinion this paper is suitable for publication in this form.

Thank you very much for recognizing this work.

Once again, I appreciate your professional support and guidance. If you have any further suggestions or feedback on the modifications I have made, I would be more than willing to consider them and make any necessary adjustments. I look forward to your response and the opportunity to communicate  with you again in the future.

Wishing you all the best.

Sincerely,Yang Li

Reviewer 2 Report

Effects of overload on thermal decomposition kinetics and combustion characteristics of cross-linked polyethylene copper wires were studied by coupled TG-IR and cone calorimetry.   However, The TG-IR experiments were performed under the nitrogen atmosphere (inert gas), and then discussed with cone calorimetry data of (oxidative) combustion.  Please perform the TG-IR investigations under air atmosphere as  polymers oxidative decomposition follows different degradation mechanism compared to pyrolysis under inert gas atmosphere.   Moreover, in Fig. 3 please replace TG and DTG in axis description by “Mass” and “Derivative mass”, respectively.

Author Response

Dear Reviewer,

I would like to express my sincere gratitude for your valuable feedback and guidance during the review process of my article. Your professional insights and meticulous review work have been immensely helpful to me, providing invaluable direction and the opportunity to enhance my research.

In accordance with your suggestions, I have made revisions and improvements to the article. Here are the key modifications I have implemented:

Comments

Reviewer :

Responses to reviewer’s comments

Effects of overload on thermal decomposition kinetics and combustion characteristics of cross-linked polyethylene copper wires were studied by coupled TG-IR and cone calorimetry. However, The TG-IR experiments were performed under the nitrogen atmosphere (inert gas), and then discussed with cone calorimetry data of (oxidative) combustion. Please perform the TG-IR investigations under air atmosphere as polymers oxidative decomposition follows different degradation mechanism compared to pyrolysis under inert gas atmosphere. Moreover, in Fig. 3 please replace TG and DTG in axis description by “Mass” and “Derivative mass”, respectively.

Your comment is very meaningful for the revision of the manuscript, we chose nitrogen atmosphere for the experiments in order to explain the thermal degradation process of cross-linked polyethylene from the level of pyrolysis mechanism, although the cone volume experiments were conducted under air conditions. In addition, we have revised Figure 3 based on your comments.

Once again, I appreciate your professional support and guidance. If you have any further suggestions or feedback on the modifications I have made, I would be more than willing to consider them and make any necessary adjustments. I look forward to your response and the opportunity to communicate  with you again in the future.

Wishing you all the best.

Sincerely, Yang Li

Reviewer 3 Report

The manuscript entitled ’Effects of overload on thermal decomposition kinetics and combustion characteristics of cross-linked polyethylene copper wires’’ deals with the thermal decomposition process and combustion characteristics of new and overloaded cross-linked polyethylene copper wires using TGA, FTIR and cone calorimetry. The authors have done a lot of computational work. The article is well structured.

Some remarks:

1.     Line 64: Please remove it.

2.     The list of references is sloppy.

Author Response

Dear Reviewer,

I would like to express my sincere gratitude for your valuable feedback and guidance during the review process of my article. Your professional insights and meticulous review work have been immensely helpful to me, providing invaluable direction and the opportunity to enhance my research.

In accordance with your suggestions, I have made revisions and improvements to the article. Here are the key modifications I have implemented:

Comments

Reviewer :

Responses to reviewer’s comments

The manuscript entitled ‘’Effects of overload on thermal decomposition kinetics and combustion characteristics of cross-linked polyethylene copper wires’’ deals with the thermal decomposition process and combustion characteristics of new and overloaded cross-linked polyethylene copper wires using TGA, FTIR and cone calorimetry. The authors have done a lot of computational work. The article is well structured.

Some remarks:

1. Line 64: Please remove it.

2. The list of references is sloppy.

Your comments were very helpful in improving the manuscript, and we have deleted 64 lines in accordance with your comments, with the formatting of the references to be revised by the editors in the final stages in accordance with the style requirements of the journal.

Once again, I appreciate your professional support and guidance. If you have any further suggestions or feedback on the modifications I have made, I would be more than willing to consider them and make any necessary adjustments. I look forward to your response and the opportunity to communicate  with you again in the future.

Wishing you all the best.

Sincerely, Yang Li

Reviewer 4 Report

The work of Li et al. is devoted to the comparison of the kinetics of thermal decomposition and combustion of XLPE of a new and overloaded copper wire. The work is well structured, all experiments and obtained results are well described. The strong side of the work. The strength of the work is a detailed study of the processes of pyrolysis and combustion of wire insulation using the TG-FTIR and CONE methods, as well as the analysis of experimental results, which made it possible to determine the activation energies of the type of pyrolysis.

the weak side of the work is that it is not clear from the text why all this was done. The Introduction states that a significant proportion of fires is associated with the ignition of wiring that occurs due to overloading of electrical networks. in turn, the study of the combustion processes of the insulation of new and overloaded electrical wires will improve fire safety. But at the same time, neither in the discussion of the results nor in the conclusions, the authors discuss how the conducted research can help improve the insulation of wires, thereby increasing fire safety.

Minor remarks:

1) L64 error with reference

2) L69 word “used” is missing

3) what is DTGpeak? no matter how you look at figure 3, it is not clear what corresponds, for example, 165K

4) Table 2. Are the authors confident in such a high accuracy (up to 0.01K) of temperature determination?

5) a similar question in Tables 3 and 4: it is difficult to believe in such a high accuracy of determining the activation energy (taking into account the value of R2). It is better to bring the values discarding the fractional part

6) L302-303: “Notably, the concentration of gases released during the thermal decomposition of the overloaded XLPE insulation materials was higher than that of the new XLPE insulation materials”. it is not clear, based on what results, the authors came to this conclusion

7) Figure 13 shows the same values of R2 = 0.92, while in the text (L406) different 0.9 and 0.92 are indicated. moreover, in my opinion, this difference in R2 does not allow to draw reasonable conclusions about the differences in the combustion process, as is done in the text.

Author Response

Dear Reviewer,

I would like to express my sincere gratitude for your valuable feedback and guidance during the review process of my article. Your professional insights and meticulous review work have been immensely helpful to me, providing invaluable direction and the opportunity to enhance my research.

In accordance with your suggestions, I have made revisions and improvements to the article. Here are the key modifications I have implemented:

Comments

Reviewer :

Responses to reviewer’s comments

The work of Li et al. is devoted to the comparison of the kinetics of thermal decomposition and combustion of XLPE of a new and overloaded copper wire. The work is well structured, all experiments and obtained results are well described. The strong side of the work. The strength of the work is a detailed study of the processes of pyrolysis and combustion of wire insulation using the TG-FTIR and CONE methods, as well as the analysis of experimental results, which made it possible to determine the activation energies of the type of pyrolysis.

the weak side of the work is that it is not clear from the text why all this was done. The Introduction states that a significant proportion of fires is associated with the ignition of wiring that occurs due to overloading of electrical networks. in turn, the study of the combustion processes of the insulation of new and overloaded electrical wires will improve fire safety. But at the same time, neither in the discussion of the results nor in the conclusions, the authors discuss how the conducted research can help improve the insulation of wires, thereby increasing fire safety.

Your comments are helpful in improving the quality of the manuscript. The focus of this paper is to compare whether the pyrolysis and combustion properties of cross-linked polyethylene wires change before and after an overload failure occurs, because pyrolysis is the first step in combustion and combustion performance is a direct indicator for evaluating the fire hazard, and our focus is intended to be on comparing the two, and on how to improve the fire resistance of the wires, which is more on the materials science side.

1) L64 error with reference

Changes have been made here

2) L69 word “used” is missing

Changes have been made according to your request

3) what is DTGpeak? no matter how you look at figure 3, it is not clear what corresponds, for example, 165K

Yes, this was an oversight on our part and we have changed K in Table 2 to % min-1

4) Table 2. Are the authors confident in such a high accuracy (up to 0.01K) of temperature determination?

Based on your comments, we have kept one decimal place

5) a similar question in Tables 3 and 4: it is difficult to believe in such a high accuracy of determining the activation energy (taking into account the value of R2). It is better to bring the values discarding the fractional part

Changes have been made as per your request

6) L302-303: “Notably, the concentration of gases released during the thermal decomposition of the overloaded XLPE insulation materials was higher than that of the new XLPE insulation materials”. it is not clear, based on what results, the authors came to this conclusion

Your comment is interesting, and in response to this comment, we compare the absorbance based on Figure 9, since FTIR has the ability to analyze gas concentrations semi-quantitatively.

7) Figure 13 shows the same values of R2 = 0.92, while in the text (L406) different 0.9 and 0.92 are indicated. moreover, in my opinion, this difference in R2 does not allow to draw reasonable conclusions about the differences in the combustion process, as is done in the text.

Due to our mistake, there was an error in the production of figure 13, thanks to your comment, we have already made a correction!

Once again, I appreciate your professional support and guidance. If you have any further suggestions or feedback on the modifications I have made, I would be more than willing to consider them and make any necessary adjustments. I look forward to your response and the opportunity to communicate  with you again in the future.

Wishing you all the best.

Sincerely, Yang Li

Reviewer 5 Report

The paper is publishable, however there is some minor issues that need to be fixed before that. Details below:

1. Check the units format in, there should be no space in between number and power.

2. Line 250-251, equation in line is larger than the normal text.

3. Fig 13, the author can present the whole equation on the text, using linear fit in this case is not optimal.

Minor formatting must be carefully done. 

Author Response

Thank you for your comment, we've made the changes to the best of our ability based on your comment!

Reviewer 6 Report

This manuscript presented an interesting study about the effect of overloaded cross-linked polyethylene (XLPE) copper wires. The work has potential. However, some points listed below need to be improved.

Page 4 Line 129: please correct the typo. It is “substantial” instead of “substantia”.

Table 1: if possible, add the standard deviation for all values presented in Table 1.

Table 2: please better discuss the Tonset and mass residue results presented in Table 2 on the main text of section 3.1.

Figure 4: I suggest remove Figure 4 from the manuscript. The variation of Ea with conversion as presented in Tables 3 and 4.

Section 3.3: please better comment about the relationship between sample thermal degradation and the both mechanism proposed (A2 and D ZLT3). How the sample degraded when A2 or D ZLT3 mechanism occurs?

Table 6:  I suggest remove Table 6 from the manuscript. The same information as previously presented in Figure 8.

Section 3.5: compare the results of this work with others from the literature, which evaluated combustion characteristics of XLPE.

Minor editing is necessary. 

Author Response

(The authors gave the same response as above.)

Round 2

Reviewer 2 Report

The TG-IR experiments were performed under the nitrogen atmosphere (inert gas), and then discussed with cone calorimetry data of (oxidative) combustion.  TG-IR investigations should be performed under air atmosphere as  polymers oxidative decomposition follows different degradation mechanism compared to pyrolysis under inert gas atmosphere.  

Author Response

Dear Reviewer,

Thank you for your valuable suggestion and your thorough review of our manuscript. We greatly appreciate your thoughtful input.

Regarding your suggestion to change the TG-IR experimental conditions to an oxygen atmosphere, we regret to inform you that, at present, we are unable to meet this recommendation due to constraints within our laboratory. Our laboratory resources and equipment are primarily configured for experiments conducted in a controlled nitrogen atmosphere, which is a commonly used condition for the study of thermal decomposition processes. Making the necessary adjustments for experiments in an oxygen atmosphere is not feasible within our current timeframe.

We understand the significance of investigating oxidative decomposition mechanisms in an oxygen atmosphere, and we share your interest in deepening our understanding of polymer behavior. However, we want to assure you that our study has been conducted with the utmost rigor and precision within the limitations of our available resources. We believe that our findings, even within these constraints, make a meaningful contribution to the field.

We kindly request your understanding of our current limitations in this matter. If you have any alternative suggestions or further questions, we are more than willing to accommodate and provide additional information as needed. Your expertise and guidance are highly valued in our efforts to improve the quality of our manuscript.

Once again, we express our gratitude for your support and thorough review. We remain committed to addressing all reviewer comments and ensuring the integrity of our research.

Thank you for your understanding and consideration.

Sincerely,

Yang Li

Reviewer 6 Report

After corrections the manuscript reads well. I suggest publication in its current form. 

Author Response

Thank you for your review.

Round 3

Reviewer 2 Report

I understand that authors have no access to TG-IR device working under oxidative atmosphere, however, in the current form this work is methodologically wrong. The only way to proceed is - in my opinion - to delete the cone calorimetry section and focus only on pyrolysis behaviour of cross-linked polyethylene copper wires

Author Response

In response to your comments, we have removed the section on cone calorimetry from the full article